# Dosing Evaluation of Ceftazidime–Avibactam in Intensive Care Unit Patients Based on Pharmacokinetic/Pharmacodynamic (PK/PD) Modeling and Simulation

**DOI:** 10.3390/antibiotics13090861

**Published:** 2024-09-09

**Authors:** Hinojal Zazo, Yuridia Aguazul, José M. Lanao

**Affiliations:** 1Area of Pharmacy and Pharmaceutical Technology, Pharmaceutical Sciences Department, University of Salamanca, 37007 Salamanca, Spain; hinojal@usal.es (H.Z.); idu031884@usal.es (Y.A.); 2Institute of Biomedical Research of Salamanca (IBSAL), 37007 Salamanca, Spain

**Keywords:** PK/PD modeling, dosing evaluation, ceftazidime–avibactam, intensive care unit, renal insufficiency, *Pseudomonas aeruginosa*

## Abstract

*P. aeruginosa* is the most common microorganism involved in many ICU-acquired infections. A correct dosage regimen is pivotal to avoiding resistance development, worse outcomes and higher mortality rates. The aim of this study was to perform a pharmacokinetic–pharmacodynamic (PK/PD) evaluation of recommended dosing regimens of ceftazidime–avibactam (CAZ–AVI) in ICU patients with different degrees of renal function for a specific strain of *Pseudomonas aeruginosa*. A semi-mechanistic PK/PD model has been developed. It allows for the simulation of CAZ–AVI steady-state plasma level curves and the evolution of bacterial growth curves. The percentage of bacterial load reduction and the value of the recommended PK/PD indices have been taken into account to define the success or failure of the regimens. Probabilistic analysis was performed using Monte Carlo simulations of two populations: control and ICU. In both populations, dosing regimens endorsed for patients with CLcr higher than 10 mL/min reach the PK/PD indices recommended, T > MIC > 90% and Cmin/MIC > 1.3. While dosage regimens endorsed for patients with CLcr of 10 mL/min or lower fail (T > MIC < 60% and Cmin/MIC < 0.35). However, proposed dosing regimens based on shortening dosing intervals for these patients would be successful, increasing bacterial load reduction by almost 50% and reaching the proposed PK/PD indices. Therefore, CAZ–AVI dosing strategies based on model-informed precision dosing (MIPD) could directly influence the efficacy of results in ICU patients with renal insufficiency.

## 1. Introduction

Optimization of antibiotic treatment based on model-informed precision dosing (MIPD) strategies allows for the calculation of initial doses in special populations supported by pharmacokinetic–pharmacodynamic (PK/PD) predictions [1].

ICU patients are a population that frequently requires antibiotic treatment and are characterized by great inter- and intra-individual pharmacokinetic variability due to the overlap of numerous pathophysiological and clinical factors. In this type of population, it is difficult to predict the outcome of exposure to standardized dosing regimens of drugs [2].

*Pseudomonas aeruginosa* (ATCC 15442) is one of the most common microorganisms involved in many intensive care unit (ICU)-acquired infections [3]. This bacterium presents a remarkable capacity to develop extensively drug-resistant (XDR) strains, so the use of a correct dosage regimen is pivotal to avoid suboptimal concentrations, which favor resistance development and are associated with worse outcomes and higher mortality rates.

One of the first-line therapies for critically ill patients infected by Gram-negative pathogens are β-lactam antibiotics. The cephalosporin ceftazidime (CAZ) is a β-lactam antibiotic with a broad spectrum of activity and low toxicity. The combination of CAZ with a non-β-lactam inhibitor, like avibactam (AVI), is active against some multidrug-resistant strains [4]. In fact, it has been shown that this combination is the best to treat infections against *P. aeruginosa* [5].

PK/PD analysis allows for the integration and joining of pharmacokinetic exposure with pharmacological effect. The quantitative relationship between PK and PD parameters is known as a PK/PD index and allows for the prediction of clinical efficacy. To correlate the exposure of these drugs with their ability to kill the target bacteria, it is widely accepted that the optimal method is to measure the time that the drug concentration remains above the bacteria’s minimal inhibitory concentration (T > MIC). For common infections, dosage regimens are designed to cover at least 40% of the T > MIC, but critically ill patients may benefit from longer and even higher values (e.g., T > 2–5 MIC) [1]. Moreover, the PK/PD index correlates with the suppression of the emergence of antibiotic resistance in *P. aeruginosa* when the target of 90% of the time above the MIC is achieved [5].

This combination, CAZ–AVI, was approved recently, in 2017 [4]. Thus, the best dosage regimen has not been established yet, particularly in special populations like ICU patients. These patients present pharmacokinetic changes of antibiotics that may alter bacterial exposure. Serum concentrations depend on pharmacokinetic (PK) parameters such as volume of distribution (V) and drug clearance (Cl), which may be modified in this type of patient due to supportive care [6]. Both CAZ and AVI elimination is dependent on renal excretion, with glomerular filtration being the main route of elimination [7]. Accordingly, although dosage will be adapted by the estimated creatinine clearance (Clcr), the achievement of PK/PD targets is pivotal to maximize effectiveness and to minimize the potential for toxicity and the development of resistance [4,8]. If not, dose reduction would be associated with a higher risk factor [9].

Based on that, the objectives of this study were to perform a pharmacokinetic–pharmacodynamic (PK/PD) evaluation using Monte Carlo simulations of recommended dosage regimens of ceftazidime–avibactam (CAZ–AVI) included in the summary of product characteristics (SmPC) [10] for a specific strain of *Pseudomonas aeruginosa*, testing for intensive care unit patients with different degrees of renal function.

## 2. Results

The PK/PD model developed allows for the simulation of drug concentrations and bacterial densities of control and ICU populations with varying degrees of renal function over the course of one week (Appendix A).

These results were validated, showing good predictive performance of the PK/PD model used. Table 1 includes the mean observed and predicted values, and the fold errors corresponding to plasma concentrations of CAZ and AVI, as well as bacterial density.

Pharmacokinetic parameters obtained for each population simulated are shown in Table 2.

In both populations, dosing regimens endorsed for renal function with a Clcr higher than 10 mL/min reached the recommended PK/PD indices (T > MIC > 90% and Cmin/MIC > 1.3) (Table 3). Regarding bacterial density, the probability of exceeding the initial inoculum at the end of the treatment is lower than 50% for all populations with Clcr higher than 10 mL/min (Appendix A). This suggests that, in patients with a Clcr lower than 10 mL/min, mainly the ICU patient population but also a high percentage of the control population, the SmPC dosage regimens would fail.

Based on previous results, maintaining the dose but with shortened dosing intervals, new dosage regimens have been suggested for control and ICU populations with a Clcr of 10 mL/min or lower. These proposed regimens have been simulated successfully (Figure 1, Figure 2, Figure 3 and Figure 4), increasing the bacterial load reduction by almost 50% (Figure 5) and reaching the PK/PD indices recommended (Table 3).

## 3. Discussion

Currently, antibiotic resistance is a serious problem, and the increasing resistance to ceftazidime and other third-generation cephalosporins represents a particularly concerning trend. The CAZ–AVI combination is one of the treatments to consider against MDR bacteria. This combination has demonstrated clinical and microbiological efficacy in a wide variety of serious infections, including those caused by MDR bacteria [5].

Understanding the pharmacokinetics of CAZ and AVI is crucial to ensuring appropriate dosing, especially in patients admitted to the ICU who may also present renal insufficiency. These patients are characterized by presenting very rapid physiological changes over time with important modifications in pharmacokinetics, both at the distribution and elimination level [2].

The PK/PD model developed in this study properly describes the time course of both AVI and CAZ, showing AFEs around 1.0 (0.5–2.5) (Table 1). There is a light over-prediction in the exposure of ICU patients because of the high variability of these patients’ vs. the few numbers of observed concentrations. Moreover, there is an over-prediction in the bacterial density with the highest doses studied. This could be due to a bias in the model for higher concentrations, although this was within the accepted limits for the AFE.

Table 2 shows the population values of the PK parameters of CAZ and AVI in the control and ICU patient populations. These parameters have been used to evaluate the different dosage regimens recommended in the SmPC using the developed PK/PD model. As seen in this table, there are important changes in the V and Cl of CAZ and AVI of ICU patients in relation to the control population. An increase in the volume of distribution is usually associated with the systemic inflammatory response syndrome (SIRS) that characterizes these patients, and modifications in clearance are dependent on the patient’s renal function [13].

Changes in the pharmacokinetics of CAZ and AVI in ICU patients significantly affect their serum levels and may compromise the antibacterial efficacy of the treatment (Appendix A).

Appendix A show the simulation of plasma levels of CAZ and AVI for the two populations, control and ICU, in patients with varying degrees of renal function. For the same populations and patients, Appendix A shows the pharmacological response, considered as the inhibition of bacterial growth for a specific strain of *Pseudomonas aeruginosa* and expressed as bacterial density (Log_10_ CFU/mL).

These results demonstrate a good pharmacological response in patients with normal renal function and moderate renal insufficiency, both in the control and ICU populations, with the different dosing regimens recommended in the SmPC. However, in patients with severe renal insufficiency (Clcr ≤ 10 mL/min), the pharmacological response worsens, especially in ICU patients where the PK/PD indices are not in line with recommendations.

According to the SmPC, the dosing interval for patients with severe renal insufficiency (Clcr lower than 10 mL/min) is much higher than for others. However, according to the obtained results, after one week of treatment, the probability of keeping or even increasing the bacterial density is high for patients in both of these populations, and especially in ICU patients, with the SmPC dosage regimens (Figure 5).

It was therefore proposed to use regimens with shortened dosing intervals due to the β-lactam antibiotic showing a time-dependent effect and the fact that the ability to kill target bacteria correlated with time. These dosing intervals should be considered for patients with a Clcr ≤ 10 mL/min, especially for ICU patients, where the likelihood of failure of the recommended treatment in the SmPC may be higher.

Figure 3 shows the probability of exceeding a bacterial density value after one week of treatment. It demonstrates that this probability is lower when the administration interval is reduced. Table 3 shows how the suggested regimens reach recommended PK/PD indices in both populations and achieve better microbiological eradication than the SmPC’s dosage regimens.

The SmPC for CAZ–AVI includes dosing recommendations for adult populations with normal renal function and those with renal impairment. However, it does not consider important changes in pharmacokinetics in ICU patients, shown in Table 2, which may influence plasma levels of ceftazidime and avibactam and their antibacterial response, as demonstrated by the results obtained in this study.

In ICU populations, changes in response could be related to a reduction in CAZ and AVI plasma concentrations, reducing the time above the MIC, as is shown in Table 3. This index is crucial for the pharmacological response, considering that cephalosporins are antibiotics whose pharmacological efficacy is time-dependent [6].

High β-lactam exposures have been associated with neurotoxicity, but safe drug concentrations were maintained in the simulated ICU populations. All maximum and minimum CAZ concentrations (Appendix A) remained below the suggested toxicity threshold (Cmax > 100 mg/L and Cmin > 35 mg/L) [1,14]. However, the minimal concentration threshold is under discussion because it is suggested that the Cmin should be above four or more times the MIC to avoid the development of resistance [5,8,15].

Antimicrobial dose optimization in severe renal insufficiency, especially ICU patients, is challenging. In addition, in clinical practice, clinicians have protocols to implement for patients with extremely low renal function. These protocols involve the use of techniques that modify the pharmacokinetic conditions of the individual, and an adaptive TDM strategy may be the most accurate approach. However, the extensive use of real-time TDM for β-lactams is still limited [11,16]. Thus, a MIPD strategy based on PK/PD modeling and simulations is a helpful tool for dosage regimen decision-making in patients with renal insufficiency, especially ICU populations, infected by *Pseudomonas aeruginosa*. The SmPC dosage regimens should also be evaluated in ICU patients infected by other bacteria.

## 4. Materials and Methods

### 4.1. Study Design

CAZ–AVI serum levels and antibacterial responses were simulated in two populations: control and ICU patients with varying degrees of renal function. The simulated control population represented patients not admitted to the ICU. Each population was also subdivided by creatinine clearance (Clcr) into six groups, with Clcr ranging from 100 mL/min to 3 mL/min. A log-normal distribution was assumed to generate the Clcr (mean for each subpopulation: 100, 60, 40, 20, 10 and 3 mL/min; CV: 20%) and body weight (Mean: 75 kg; CV: 13%).

Using the PK/PD model developed, dosing regimens recommended in the drug’s SmPC for patients with different degrees of renal function were evaluated (Table 4).

### 4.2. Pharmacokinetic–Pharmacodynamic (PK/PD) Modeling

A specific semi-mechanistic CAZ–AVI PK/PD model was developed and used to simultaneously simulate drug serum levels and antibacterial responses in control and ICU populations with varying degrees of renal function for several dosage regimens shown in the SmPC of CAZ–AVI [10].

Pharmacokinetic model

A population PK model was used to define the pharmacokinetic behaviour of CAZ–AVI. It is a one-compartment kinetic model with linear elimination and a correlation between plasma clearance and renal function.

Equations (1) and (2) estimated the maximum and minimum drug concentrations in the steady state.
(1)$$C_{max}^{ss}\;(\mathrm{mg}/L)=\frac{Ko}{CL}*\;\frac{\left(1-e^{-\frac{Cl}{V}T}\right)}{\left(1-e^{-\frac{Cl}{V}\tau }\right)}$$
(2)$$C_{min}^{ss}\;(\mathrm{mg}/L)=\frac{Ko}{CL}*\;\frac{\left(1-e^{-\frac{Cl}{V}T}\right)}{\left(1-e^{-\frac{Cl}{V}\tau }\right)}*e^{-\frac{Cl}{V}(\tau -T)}$$
where Ko is the infusion rate constant (dose/T), CL is the serum clearance, T is time of infusion and τ is the dosing interval.

Equations (3) and (4) estimated the distribution volume (V (L)) and serum clearance (CL (L/h)) of the control population and ICU population using PK parameters (Table 5) from the literature [17,18,19].
V (L) = Vd x Weight (kg) (3)
Cl (L/h) = Cli + (CLs x Clcr (mL/min))(4)
where Vd is the value of the distribution volume coefficient in steady state, Cli is the value of the ordinate at the origin, and CLs is the slope of the correlation between drug clearance and creatinine clearance (Clcr (mL/min)). The estimation errors for clearance parameters (Cli and Cls) and interindividual variability for vistribution volume coeficient (Vd) are specified in Table 5.

Pharmacodynamic model

The PD model consider the rate of change in the simulated bacterial density (Log_10_ CFU/mL) of a specific strain (2154) of *Pseudomonas aeruginosa* population over time. It is expressed as a function of the kill rate caused in response to CAZ–AVI concentrations based on the published mathematical model developed by Sy et al. [12]. It describes the time course of a bacterial load [N(t)] as a function of different concentrations of a antimicrobial agent [C(t)] and also takes into account how the degradation of CAZ influences bacterial density-dependent concentrations and AVI concentrations.

Two bacterial populations were used in this study:-Population 1 (P₁) with active microbial growth and an initial inoculum of 10⁶ CFU/mL.-Population 2 (P₂) in the resting phase with an initial inoculum of 1/10⁷ CFU/mL.

The developed PD model allows us to simulate the evolution of bacterial growth, taking into account the inhibitory effects of both drugs. Additionally, degradation by hydrolysis of ceftazidime caused by bacteria and influenced by the concentration of avibactam was described.

The dynamic growth model was defined according to the following equations:(5)$$\frac{d{log}_{10}\left(P_{1}\right)}{dt}=\left[I_{1}k_{growth,1}\left(1-\frac{{log}_{10}(P_{1}+P_{2)}}{{log}_{10}N_{max}}\right)\right]{log}_{10}\left(P_{1}\right)-\left[I_{2}k_{death,1}+k_{1-2}\right]{log}_{10}\left(P_{1}\right)$$
where: I_1_ represents a growth retardation function; I_2_ represents a death function; k_death_,_1_, (h^−1^) is the first-order constant of bacterial death and is dependent on the concentration of ceftazidime for the destruction of cells in the P1; k_growth_,_1_, (h^−1^) is the first-order bacterial growth constant associated with the log_10_ of the active population P_1_; k_1–2_, log_10_ (CFU/mL/h) is the rate constant for the conversion of bacterial cells from the state of P_1_ to P_2_, regardless of the concentrations of CAZ or AVI; N_max_ is the maximum load capacity achievable in the system; P_1_ (CFU/mL) is the number of cells in active growth per unit of volume at time t(h), and P_2_ (CFU/mL) is the number of bacteria per unit of volume in the resting state.

Likewise, k_death_,_1_ is defined by the following equations:(6)$$k_{death,1}=\frac{E_{max}\cdot {CAZ}^{\gamma }}{(Aexp\left(-\alpha \cdot AVI\right)+Bexp(-\beta \cdot AVI){)}^{\gamma }+{CAZ}^{\gamma }}$$
and
(7)$$\frac{d\left(P_{2}\right)}{dt}=\left[k_{growth,2}\left(1-\frac{{log}_{10}(P_{1}+P_{2)}}{{log}_{10}N_{max}}\right)\right]\left(P_{2}\right)+k_{1-2}\left(P_{1}\right)$$
where A (mg/L) is a coefficient of the exponential function to characterize the EC50 of ceftazidime; AVI is the concentration of avibactam; B (mg/L) is another coefficient of the exponential function; CAZ is the concentration of ceftazidime; E_max_ (h^−1^) is the maximum constant of the bacterial death rate due to ceftazidime; E_max_ (h^−1^) is the first-order bacterial growth constant associated with the log_10_ of the active population P₂; α (L/mg) is the constant associated with parameter A that describes the relationship between the concentration of AVI and the potency of CAZ; β (L/mg) is the constant associated with parameter B that describes the relationship between the concentration of avibactam and the potency of ceftazidime; and ϒ is the Hill coefficient that characterizes the slope of the sigmoidal E_max_ curve associated with the increase in the potency of ceftazidime by avibactam.

The variable I represents a delay function, which describes the effect of the lowest concentration of ceftazidime to delay the bacterial growth of population P_1_:I_i = {1, if ceftazidime = 0 (all conditions); otherwise, 1 − exp (−δi x t)}(8)
where δi (h^− 1^) is the exponential delay constant; the subscript i is 1 or 2, of which 1 represents the growth delay functions and 2 represents the death delay functions, respectively.

The following equation was used to model the bacterial impact on ceftazidime degradation.
(9)$$\frac{dCAZ}{dt}=-\frac{{Deg}_{max}\cdot P_{1}^{\varphi }}{K_{m}^{\varphi }+P_{1}^{\varphi }}\left(1-\frac{AVI}{{IC}_{50}+AVI}\right)CAZ$$
where Deg_max_ is the maximum degradation rate constant of ceftazidime; IC₅₀ is the concentration of avibactam that produces a 50% decrease in the degradation rate; Km is the log_10_ transformed CFU number density that produces 50% of the maximum degradation rate; and φ is the Hill coefficient that characterizes the slope of the Hill function (46). The values of the parameters that were used in the simulation of the pharmacodynamic model are shown in Table 6.

The reduction in bacterial density (Log_10_ CFU/mL) after one week of treatment was taken into account to define the success or failure of the dosage regimens. Although, in clinical microbiological eradication, failure was defined as the presence of the same bacterial pathogen in blood or body fluid cultures ≥7 days after initiation of the CAZ–AVI treatment [20].

Model validation

The predictive performance of the PK model was assessed by a numerical predicted check (NPC). The observed PK and PD data from the literature [7,11,12] overlaid the simulated plasma concentration time profiles of CAZ and AVI, as well as the simulated bacterial density (log_10_CFU/mL). A total of 1000 virtual patients from each subpopulation were considered to calculate prediction intervals (PI) of 90%.

If observed concentrations from the literature [7,11,12] were distributed within the 90% PI, the model prediction capability was deemed to be adequate [21].

The overall predictability of the model was evaluated based on the average fold error (AFE). The calculated AFE was considered acceptable in the literature if it was within a 2-fold error (0.5–2-fold) [22]. The equation used for the calculation of AFE was the following:(10)$$AFE=10^{\frac{1}{n}\sum log(\frac{{PRED}_{Mean}}{OBS})}$$

Monte Carlo simulations

Probabilistic analysis was performed using Monte Carlo simulations based on 1000 runs for each analysis during the first week of treatment. The geometric mean and the geometric standard deviation of the PK/PD parameter values were employed. The parameters were sampled from a log-normal distribution. For the simulations, the software package GoldSim Pro v.10.5 (GoldSim Technology Group, Issaquah, WA, USA) was used.

### 4.3. PK/PD Analysis

The PK/PD model developed allowed for the simulation of steady-state plasma level curves, the evolution of bacterial growth curves, and PK/PD efficacy indices. To evaluate the success or failure of the dosage regimens, the reduction in bacterial load after one week of treatment and the value of the recommended PK/PD indices were taken into account.

Cephalosporins exert a time-dependent bactericidal effect. Therefore, the recommended efficacy indices are the following [6]:-Time during the serum drug concentration remains above the minimum inhibitory concentration (MIC) (T > MIC)-Ratio of the trough serum concentration to the MIC (Cmin/MIC).

The MIC assumed in this model was the MIC_50_ for Meropenem-non-susceptible *P. aeruginosa*, one of the most common strains in the ICU. The values for CAZ and AVI were 4 and 1 mg/L, respectively [23]. The PK/PD indices’ values considered that significant predictors of a positive clinical outcome were T > MIC > 90% [5] and Cmin/MIC > 1.3, which were the PK/PD targets [24].

## 5. Conclusions

The likelihood of failure of recommended CAZ–AVI treatments may be higher in patients with a Clcr lower than 10 mL/min, especially ICU patients, according to the PK/PD model tested. Thus, in these populations, regimens that reduce dosing intervals could potentially be successful against *Pseudomonas aeruginosa* infections. Therefore, CAZ–AVI dosing strategies could directly influence the efficacy of results in ICU patients. Although PK/PD modeling and simulation studies are useful for planning dosing strategies—mainly in special populations, such as ICU patients—subsequent clinical evaluation accompanied by therapeutic drug monitoring (TDM) strategies, is recommended.

## Figures and Tables

**Figure 1 antibiotics-13-00861-f001:**
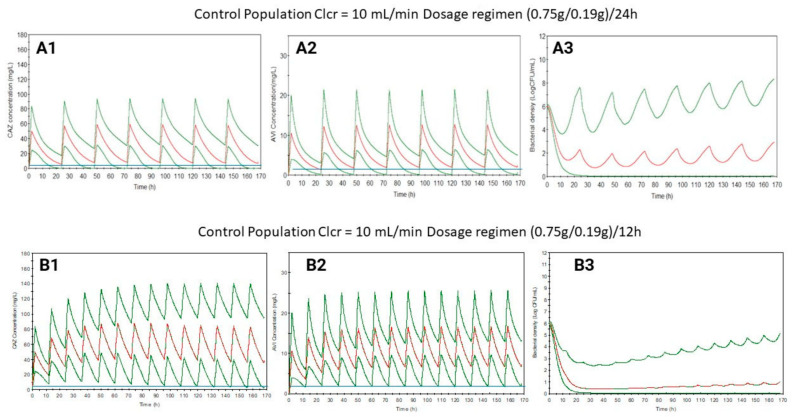
Ceftazidime (**1**) and avibactam (**2**) plasma level curves, and bacterial growth (**3**) levels simulated for control patients with Clcr = 10 mL/min using the dosage regimen from the SmPC ((0.75/0.19)/24) (**A**) or the suggested regimen ((0.75/0.19)/12) (**B**). Blue line: MIC value (4 mg/L for CAZ and 1 mg/L for AVI). Red line: mean. Green line: 5/95%.

**Figure 2 antibiotics-13-00861-f002:**
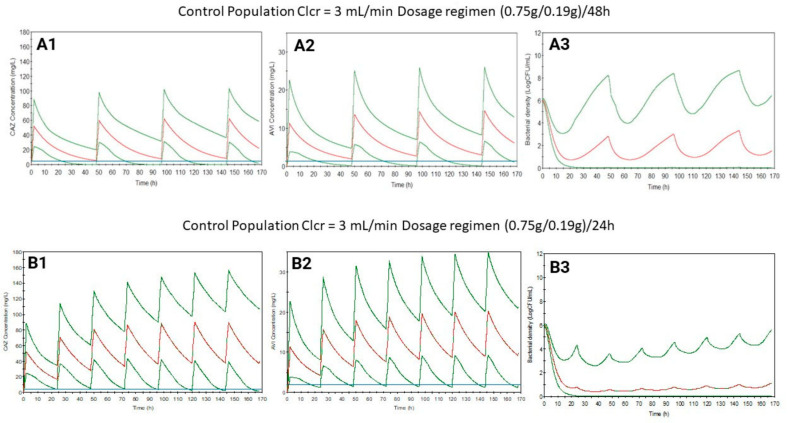
Ceftazidime (**1**) and avibactam (**2**) plasma level curves, and bacterial growth (**3**) levels simulated for control patients with Clcr = 3 mL/min using the dosage regimen from the SmPC ((0.75/0.19)/48) (**A**) or the suggested regimen ((0.75/0.19)/24) (**B**). Blue line: MIC value (4 mg/L for CAZ and 1 mg/L for AVI). Red line: mean. Green line: 5/95%.

**Figure 3 antibiotics-13-00861-f003:**
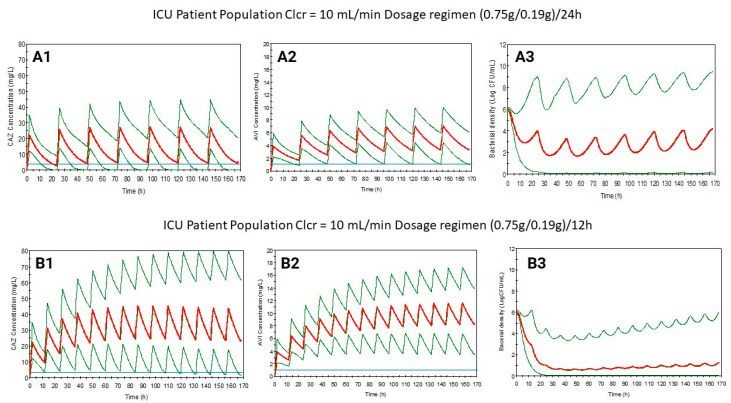
Ceftazidime (**1**) and avibactam (**2**) plasma level curves, and bacterial growth (**3**) levels simulated for ICU patients with Clcr = 10 mL/min using the dosage regimen from the SmPC ((0.75/0.19)/24) (**A**) or the suggested regimen ((0.75/0.19)/12) (**B**). Blue line: MIC value (4 mg/L for CAZ and 1 mg/L for AVI). Red line: mean. Green line: 5/95.

**Figure 4 antibiotics-13-00861-f004:**
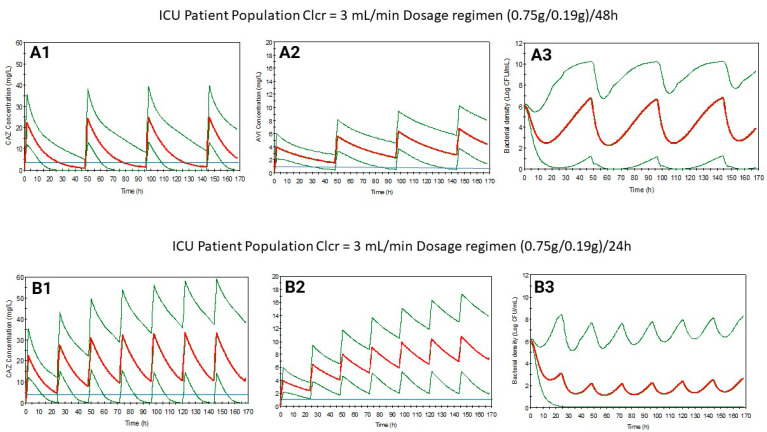
Ceftazidime (**1**) and avibactam (**2**) plasma level curves, and bacterial growth (**3**) levels simulated for ICU patients with Clcr = 3 mL/min using the dosage regimen from the SmPC ((0.75/0.19)/48) (**A**) or the suggested regimen ((0.75/0.19)/24) (**B**). Blue line: MIC value (4 mg/L for CAZ and 1 mg/L for AVI). Red line: mean. Green line: 5/95.

**Figure 5 antibiotics-13-00861-f005:**
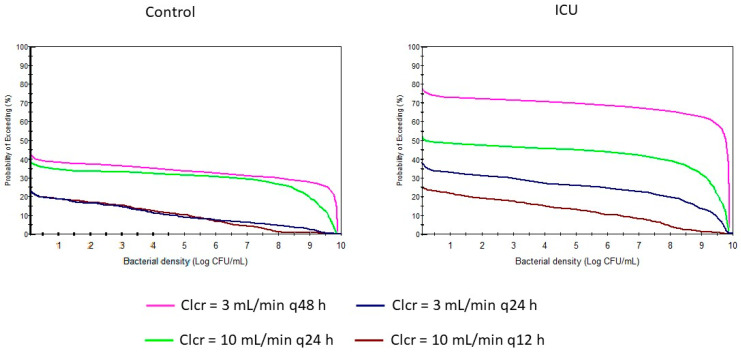
Probability of exceeding a bacterial density value in different simulated dosage regimens for ICU patients with renal insufficiency. Green line: Clcr of 10 mL/min with a dosing interval of 24 h; pink line: Clcr of 3 mL/min with a dosing interval of 48 h; dark blue line: Clcr of 10 mL/min with a dosing interval of 12 h; brown line: Clcr of 3 mL/min with a dosing interval of 24 h.

**Table 1 antibiotics-13-00861-t001:** Mean simulated values and predictability of the model (based on the average fold error).

	Mean Value Predicted	Mean Value Observed	AFE
${CAZ}_{max}^{ss}$ ^a^ (mg/L)	95.9	90.1 [7]	1.06
${CAZ}_{max}^{ss}$ ^b^ (mg/L)	67.8	53.8 [11]	1.27
${AVI}_{max}^{ss}$ ^c^ (mg/L)	15.9	14.5 [7]	1.10
Bacterial density ^d^ (LogCFU/mL)	9.23	9.1 [12]	1.01
Bacterial density ^e^ (LogCFU/mL)	8.78	8.0 [12]	1.10
Bacterial density ^f^ (LogCFU/mL)	6.57	4.7 [12]	1.40

${AVI}_{max}^{ss}$: avibactam concentration at a steady state; ${CAZ}_{max}^{ss}$ ${AVI}_{max}^{ss}$: ceftazidime concentration at steady state; AFE: average-fold error. ^a^ Concentrations in the control population with Clcr of 100 mL/min. ^b^ Concentrations in the ICU patient population with Clcr of 100 mL/min. ^c^ Concentration s in the control population with Clcr of 100 mL/min. ^d^ Concentrations used in the static time–kill in vitro study: 2 mg CAZ/L + 4 mg AVI/L. ^e^ Concentrations used in the static time–kill in vitro study: 4 mg CAZ/L + 4 mg AVI/L. ^f^ Concentrations used in the static time–kill in vitro study: 8 mg CAZ/L + 4 mg AVI/L.

**Table 2 antibiotics-13-00861-t002:** Population PK parameters of ceftazidime–avibactam in patients with different degrees of renal function for the two types of simulated populations.

Population	Clcr (mL/min)	Ceftazidime PK Parameters			Avibactam PK Parameters		
		Cl (L/h) [CV (%)]	V (L) [CV (%)]	t ½ (h) [CV (%)]	Cl (L/h) [CV (%)]	V (L) [CV (%)]	t ½ (h) [CV (%)]
Control	100	6.08 [35.2]	15.7 [40.8]	1.99 [53.3]	12.2 [26.5]	20.2 [53.6]	1.22 [59.0]
	60	3.82 [34.5]		3.17 [53.0]	7.57 [26.6]		1.96 [59.2]
	40	2.26 [40.3]		4.55 [52.7]	5.19 [26.8]		2.87 [59.2]
	20	1.59 [34.0]		7.61 [52.7]	2.99 [28.1]		5.00 [59.8]
	10	0.93 [35.5]		13.1 [54.2]	1.65 [32.1]		9.24 [62.6]
	3	0.48 [47.9]		27.7 [65.3]	0.73 [53.4]		24.4 [80.1]
ICU	100	5.44 [31.8]	34.5 [33.9]	4.84 [47.7]	10.9 [28.0]	51.0 [31.7]	3.49 [41.5]
	60	3.73 [31.4]		7.04 [47.3]	6.90 [27.7]		5.49 [41.02]
	40	2.86 [31.8]		9.23 [47.7]	4.86 [28.0]		7.80 [41.3]
	20	2.05 [35.1]		13.1 [49.3]	2.99 [32.8]		12.8 [42]
	10	1.56 [41.7]		17.9 [53.4]	1.85 [36.2]		21.4 [45.3]
	3	1.22 [50.8]		24.6 [61.7]	1.06 [55.7]		43.2 [60.5]

CAZ: ceftazidime; AVI: avibactam; Cl: total clearance; Clcr: creatinine clearance, CV: coefficient of variation, V: distribution volume.

**Table 3 antibiotics-13-00861-t003:** PK/PD indices and bacterial density values obtained with the developed PK/PD model.

Dosage Regimen	Population	Clcr (mL/min)	Dosage Regimen ((CAZg/AVIg)/h)	Bacterial Density Change ^a^	T > MIC	Cmin/MIC
				(%) (5/95%)	(%) (5/95%)	Mean (5/95%)
Summary of Product Characteristics	Control	100	(2.0/0.5)/8	−63.7 (−99.3/24.5)	95.4 (62.9/100)	0.81 (0.00/4.75)
		60	(2.0/0.5)/8	−78.3 (−99.3/−1.50)	100 (75.9/100)	4.42 (0.03/15.4)
		40	(1.0/0.25)/8	−76.0 (−99.3/3.83)	100 (74.5/100)	4.25 (0.12/12.9)
		20	(0.75/0.19)/12	−69.5 (−99.3/16.0)	100 (66.8/100)	3.40 (0.06/10.7)
		10	(0.75/0.19)/24	−51.2 (−99.0/38.5)	100 (53.1/100)	2.04 (0.01/7.62)
		3	(0.75/0.19)/48	−44.7 (−98.5/44.0)	100 (42.5/100)	2.08 (0.00/9.13)
	ICU	100	(2.0/0.5)/8	−77.5 (−99.3/0.67)	100 (77.6/100)	4.47 (0.20/12.7)
		60	(2.0/0.5)/8	−84.8 (−99.3/−19)	100 (100/100)	9.76 (1.70/21.6)
		40	(1.0/0.25)/8	−80.0 (−99.3/−5.33)	100 (98.4/100)	6.43 (1.06/14.3)
		20	(0.75/0.19)/12	−65.7 (−99.3/25.5)	100 (64.6/100)	3.50 (0.00/19.0)
		10	(0.75/0.19)/24	−30.2 (−96.83/57.83)	100 (38.2/100)	1.18 (0.00/19.0)
		3	(0.75/0.19)/48	13.0 (−78.8/70.5)	59.6 (23.5/100)	0.35 (0.00/19.0)
Suggested	Control	10	(0.75/0.19)/12	−83.7 (−99.3/−15.3)	100 (100/100)	10.1 (1.88/22.0)
		3	(0.75/0.19)/24	−80.8 (−99.3/−6.5)	100 (75.0/100)	9.37 (0.53/25.7)
	ICU	10	(0.75/0.19)/12	−79.5 (−99.3/−0.83)	100 (94.1/100)	6.00 (0.97/13.2)
		3	(0.75/0.19)/24	−56.5 (−99.2/37)	100 (44.8/100)	2.63 (0.01/9.35)

AVI: avibactam; Clcr: creatinine clearance; Cmin: trough concentration; MIC: minimum inhibitory concentration; T > MIC: time during which the serum drug concentration remains above the minimum inhibitory concentration. ^a^ Bacterial density change from time 0 and an initial inoculum of 6 Log_10_ CFU/mL to after one week of treatment.

**Table 4 antibiotics-13-00861-t004:** Dosage regimens of CAZ–AVI evaluated in control and ICU patient populations.

	Clcr (mL/min)	CAZ Dose (g)	AVI Dose (g)	Interval (h)
SmPC	100	2.0	0.5	8
	60	2.0	0.5	8
	40	1.0	0.25	8
	20	0.75	0.19	12
	10	0.75	0.19	24
	3	0.75	0.19	48

AVI: avibactam, CAZ: ceftazidime. IV administration by infusion for 2 h.

**Table 5 antibiotics-13-00861-t005:** Values of the PK parameters for control and ICU patient populations.

Parameter	Control Population		ICU Patient Population	
	CAZ	AVI	CAZ	AVI
Vd (L/kg)	0.21 (38.1)	0.27 (51.8)	0.46 (30.4)	0.68 (27.9)
Cli (L/h)	0.39 (55.6)	0.53 (65.6)	1.15 (54.8)	0.89 (65.2)
CLs	0.06 (36.9)	0.12 (27.3)	0.04 (37.2)	0.10 (30.0)

CAZ: ceftazidime; AVI: avibactam; Vd: distribution volume coefficient; Cli: clearance intercept term; CLs: clearance slope term. Mean (CV (%)).

**Table 6 antibiotics-13-00861-t006:** Parameter values for the dynamic bacterial population growth model. Adapted from Sy et al. [12].

Parameter	Description	Value [CV (%)]	Units
N_max_	Maximum load capacity achievable in the system.	9.89 [2.06]	CFU/mL
k_growth,1_	Bacterial growth associated with the log_10_ of the active population P_1_.	0.346 [20.9]	h^−1^
k_growth,2_	Bacterial growth rate constant associated with the log_10_ of the active population P₂.	(1/10^7^) × K_growth,1_	h^−1^
E_max_	Maximum kill rate constant due to CAZ.	0.240 [16.0]	h^−1^
A	First coefficient of the biexponential function to characterize the EC50 of CAZ.	52.3 [17.2]	mg/L
B	Second coefficient of the biexponential function to characterize the EC50 of CAZ.	12.6 [26.0]	mg/L
α	Exponential constant associated with parameter A that describes the relationship between the concentration of AVI and the potency of CAZ.	2.38 [119]	L/mg
β	Exponential constant associated with parameter B that describes the relationship between the concentration of AVI and the potency of CAZ.	9.67 E^2^ [7.01]	L/mg
ϒ	Hill coefficient characterizing the steepness of the slope of the sigmoidal Emax curve associated with the increase in potency of CAZ by AVI.	2.60 [34.23]	-
δ_1_	Exponential constant of the delay function to retard the active population, P₁.	4.23 E^2^ (fixed)	h^−1^
δ_2_	Exponential constant of the delay function to slow the initial death of the active population, P₁.	0.213 [17.46]	h^−1^
k_1–2_	Rate constant for the conversion of bacterial cells from active to resting states.	5.0 E^3^ (fixed)	CFU/mL/h
Deg_max_	Maximum degradation rate constant of CAZ.	7.71 E^2^ [51.9]	h^−1^
*K* _m_	CFU density that yielded 50% of the maximum degradation rate.	8.5 (fixed)	CFU/mL
φ	Hill coefficient that characterizes the slope of the sigmoid Emax model for CAZ degradation.	1.46 [82.2]	-
IC₅₀	AVI concentration that yielded a 50% decrease in the degradation rate.	1.96 [58.2]	mg/L

AVI: avibactam; CAZ: ceftazidime.

## Data Availability

Dataset is avialable in the European Repository ZENODO into the IBSAL Community, DOI: 10.5281/zenodo.13731674.

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
