# Peer review of "Dosing Evaluation of Ceftazidime–Avibactam in Intensive Care Unit Patients Based on Pharmacokinetic/Pharmacodynamic (PK/PD) Modeling and Simulation"

_antibiotics, 2024, doi:10.3390/antibiotics13090861_

Round 1

Reviewer 1 Report

Comments and Suggestions for Authors

This manuscript reported a newly developed semi-mechanistic PK/PD model which allows the simulation of CAZ/AVI steady state plasma level curves and the evolution of bacterial growth curves. 

Its overall well presented and clearly illustrated. The PK/PD model can be a good application for the pre-clinical evaluation of novel antibiotics.

Author Response

Thank you for your comments. 

Reviewer 2 Report

Comments and Suggestions for Authors

Dear Authors,

I congratulate you for your extensive work entitled " Dosing evaluation of ceftazidime/avibactam in intensive care unit patients based on pharmacokinetic-pharmacodynamic (PK/PD) modeling and simulation."

The manuscript titled " Dosing evaluation of ceftazidime/avibactam in intensive care unit patients based on pharmacokinetic-pharmacodynamic (PK/PD) modeling and simulation." presents a significant contribution to the Drug Disposition Research field. The study investigates pharmacokinetic/pharmacodynamic (PK/PD) evaluation of recommended dosing regimens of ceftazidime/avibactam (CAZ/AVI) in ICU patients with different degrees of renal function for a specific strain of Pseudomonas aeruginosa.

Specific comments:
The manuscript has been well written and needs no further changes except for the below comments:

Please add ATCC strain code for P. aeruginosa.

Please mention the PK modeling equation in the supplementary section.

Please explain the significance of efficacy indices for readers.

Explain MIC50 and MIC90 for CAZ and AVI against P. aeruginosa

By addressing these points, the manuscript could be significantly strengthened, enhancing its contribution to Nano Drug Delivery.

Author Response

Thank you for your comments, following are the answers to the specific comments:
1.- Please add ATCC strain code for P. aeruginosa.

The code has been included in the introduction, line 39.

2.- Please mention the PK modeling equation in the supplementary section.

Due to all PK/PD modelling equations being in Material and Method section, these equations (Equations 1 and 2) have been added there (lines 238-241).

3.- Please explain the significance of efficacy indices for readers.

The explanation has been included in the introduction, lines 51 to 53.

4.- Explain MIC50 and MIC90 for CAZ and AVI against P. aeruginosa

MIC value used corresponds to MIC50 for Meropenem-non-susceptible P. aerugionosa. This explanation has been added to the manuscript (lines 350-351).

Reviewer 3 Report

Comments and Suggestions for Authors

In the manuscript, the authors have constructed a PK/PD model to predict dosing of ceftazidime and avibactam combination to treat P. aeruginosa infections in ICU patients with renal insufficiency. The study is interesting and has the potential to improve efficacy of antibiotic therapy. The manuscript is well-structured and well-written. The introduction and discussion sections include necessary information. I recommend that the manuscript be considered for publication.  

Author Response

Thank you for your comments.  

Reviewer 4 Report

Comments and Suggestions for Authors

The authors have conducted an analysis evaluating alternative dosing regimens in ICU patients with P. aureginosa infection based on the existing information of the PK/PD properties of AVI/CAZ in adult patients. Overall, the article is well written and interesting. Some minor issues require further clarification by the authors: 

1. The validation of the population PK/PD model is unclear. The authors referred to a VPC, but the figure is not included in the manuscript nor in the supplementary material. Only a numerical predictive check (NPC) has been provided in Table 1, but the adequacy of the popPK/PD model to describe the time-course of both AVI/CAZ is required. 

2. Based on the AFE results, the population PKPD model tends to systematically over-predict the exposure and bacterial density, specially in ICU patients. Is there any rational explanation for this bias?

3. Figure 5 should state "ICU", please correct

4. Equation 4 has two "exp" terms for the A parameter, please correct

5. Figures 1-4 depicting the AVI and CAZ should be the y-axis in log-scale, please correct

6. The strategy for the dose optimization is partially unclear, since no information regarding the pre-established PK and PD targets for the Monte-Carlo simulation were included in the Material and Method section. This information should be included in the manuscript in order to clarify the numerical criteria. 

7. Have the authors evaluated alternative dosing regimens through an optimal design approach? What is the rationale for selecting alternative dose levels and/or posologies? 

Author Response

Reviewer 4 Thank you for your comments, following are the answers to the minor issues require: 

  1. The validation of the population PK/PD model is unclear. The authors referred to a VPC, but the figure is not included in the manuscript nor in the supplementary material. Only a numerical predictive check (NPC) has been provided in Table 1, but the adequacy of the popPK/PD model to describe the time-course of both AVI/CAZ is required. 

Thank you for your observation, there was an error, and the predictive performance was assessed by the NPC. It has been corrected in the manuscript (line 325).

Due to the low number of patients studied, the VPC was not included in the manuscript nor in the supplementary material.

  1. Based on the AFE results, the population PKPD model tends to systematically over-predict the exposure and bacterial density, specially in ICU patients. Is there any rational explanation for this bias?

The PK/PD model developed describes properly the time-course of both AVI/CAZ, AFE around 1.0 when limits are 0.5-2.5 (Table 1). There is a lightly over-prediction in the exposure of ICU patients which could be because of the high variability of these patients vs the few number of patients studied. Moreover, there is also an overprediction in the bacterial density with the highest doses studied. This could be due to a bias of the model for the higher concentrations, although within the limits accepted for the AFE (lines 149-154).

  1. Figure 5 should state "ICU", please correct

Thank you for your observation, the figure has been corrected.

  1. Equation 4 has two "exp" terms for the A parameter, please correct

Thank you for your observation, the equation has been corrected.

  1. Figures 1-4 depicting the AVI and CAZ should be the y-axis in log-scale, please correct

CAZ and AVI concentrations are in decimal scale to be more understable for reader. Previously, we also think about the use of log-scale in the y-axis. However, with the IP of 90%, decimal scale is more clear as you can see in the following figures.

Figure A1 of the manuscript in decimal scale

Figure A1 of the manuscript in log-scale

  1. The strategy for the dose optimization is partially unclear, since no information regarding the pre-established PK and PD targets for the Monte-Carlo simulation were included in the Material and Method section. This information should be included in the manuscript in order to clarify the numerical criteria. 

The PK/PD targets are included in Material and Methods section Point 4.3. PK/PD analysis (Lines 357-359).

  1. Have the authors evaluated alternative dosing regimens through an optimal design approach? What is the rationale for selecting alternative dose levels and/or posologies? 

Although the reviewer's suggestion to use an optimal design approach for dosing regimens is interesting, in this case it has not been carried out because the aim was to evaluate the regimens proposed in the SmPC through PK/PD models, proposing small changes that would allow improving the pharmacological efficacy of the dosing regimens proposed in the SmPC. A shortening dosing interval was selected because β-lactam antibiotic ability to kill the target bacteria correlates with the time. This explanation has been included in the discussion, lines 178-183.
